# The OmpR-like Transcription Factor as a Negative Regulator of *hrpR/S* in *Pseudomonas syringae* pv. *actinidiae*

**DOI:** 10.3390/ijms232012306

**Published:** 2022-10-14

**Authors:** Fu Zhao, Taihui Zhi, Renjian Hu, Rong Fan, Youhua Long, Fenghua Tian, Zhibo Zhao

**Affiliations:** 1Department of Plant Pathology, College of Agriculture, Guizhou University, Guiyang 550025, China; 2Kiwifruit Engineering & Technology Research Center, Guizhou University, Guiyang 550025, China

**Keywords:** *Pseudomonas syringae* pv. *actinidiae*, type III secretion system, bacterial canker of kiwifruit, *hrpR/S*, transcription regulation

## Abstract

Bacterial canker of kiwifruit is a devastating disease caused by *Pseudomonas syringae* pv. *actinidiae* (*Psa*). The type III secretion system (T3SS), which translocates effectors into plant cells to subvert plant immunity and promote extracellular bacterial growth, is required for *Psa* virulence. Despite that the “HrpR/S-HrpL” cascade that sophisticatedly regulates the expression of T3SS and effectors has been well documented, the transcriptional regulators of *hrpR/S* remain to be determined. In this study, the OmpR-like transcription factor, previously identified by DNA pull-down assay, was found to be involved in the regulation of *hrpR/S* genes, and its regulatory mechanisms and other functions in *Psa* were explored through techniques including gene knockout and overexpression, ChIP-seq, and RNA-seq. The OmpR-like transcription factor had binding sites in the promoter region of the *hrpR/S*, and the transcriptional level of the *hrpR/S* increased after the deletion of *OmpR-like* and decreased upon its overexpression in an *OmpR-like* deletion background. Additionally, *OmpR-like* overexpression reduced the strain’s capacity to form biofilms and lipopolysaccharides, led to its slow growth in King’s B medium, and reduced its swimming ability, although there was no significant effect on its pathogenicity against kiwifruit hosts. Our results indicated that OmpR-like directly and negatively regulates the transcription of *hrpR/S* and may be involved in the regulation of multiple biological processes in *Psa*. Our results provide a basis for further understanding the transcriptional regulation mechanism of *hrpR/S* in *Psa*.

## 1. Introduction

The bacterial canker of kiwifruit, caused by *P. syringae* pv. *actinidiae* (*Psa*), has become the most devastating disease in its cultivation, threatening the development of the kiwifruit industry [1]. Based on multi-locus sequence analysis and phylogenomic analysis, *Psa* is currently classified into five biovars, namely biovars 1, 2, 3, 5, and 6, with biovar 3 having contributed to the worldwide pandemic [2,3,4]. Compared to *Psa*1 and *Psa*2, *Psa*3 displays a strong activation of all hypersensitive reaction and pathogenicity (*hrp*) and *hrp* conserved (*hrc*) cluster genes, which encode components of the type III secretion system required for bacterial pathogenicity and are involved in responses to environmental signals [5].

The type III secretion system (T3SS) is key to pathogenic virulence. It is a syringe-like apparatus located on the bacterial cell wall and membrane, which inhibits host defense-related responses by delivering effector proteins into host cells [6,7]. A key regulator of the T3SS in *P. syringae* is HrpL, a member of the ECF family of alternative sigma factors that binds a highly conserved *hrp* box sequence within the promoters of T3SS-associated genes [8,9]. Expression of *hrpL* is induced by HrpR and HrpS, two NtrC-like enhancer-binding proteins that bind the *hrpL* promoter and recruit the sigma factor RpoN (σ^54^), are encoded by a single *hrpRS* operon, and function as a heterohexamer [9,10,11]. This is known as the HrpR/S-HrpL-T3SS cascade, and there are many other intricate regulatory mechanisms of the T3SS [12,13,14]. Furthermore, both HrpR and HrpS proteins lack receptor domains found in most NtrC family proteins, suggesting that they are not directly regulated by cellular signaling [9]. T3SS genes are induced when bacteria are in plant tissues or media, mimicking the conditions found in the apoplast [15,16,17]. The expression of the T3SS genes is stimulated at low osmolarity and acidic minimal media that contain carbon sources such as fructose and is further induced in the presence of iron [18,19,20]. In contrast, T3SS genes are repressed in most rich media that contain complex nitrogen sources or broad-spectrum amino acid sources, suggesting that the T3SS is not only controlled by endogenous factors, but also responds to external signals [18,19].

Transcriptional regulation mechanisms of the “HrpR/S-HrpL-T3SS” cascade have been described in many studies. Besides the regulation of AlgU and HrpA, the induction of the *hrpRS* operon is further regulated by the two-component systems (TCSs) RhpRS, CvsRS, and GacAS [21,22,23,24]. RhpRS is the main TCS regulating T3SS, and the *rhpR* and *rhpS* genes are located in an operon, encoding a response regulator and a cognate sensor kinase, respectively. Phosphorylated RhpR inhibits T3SS gene expression, while RhpS reverses the phosphorylation of RhpR under T3SS-inducing conditions [25,26,27]. Although HrpR and HrpS regulate the transcription of *hrpL* gene as a heterohexamer, they also act individually: HrpR can act by binding to nucleotides, while HrpS interacts with other intracellular proteins such as HrpV [11]. HrpV directly binds the HrpS AAA+ domain, thereby interfering with HrpR/S to form heterohexamer and inhibiting the expression of the T3SS gene, while HrpG removes this inhibition by binding to HrpV [11,28]. In addition to being regulated by HrpR/S, *hrpL* expression is also subject to negative autogenous control mediated via HrpL binding at the *hrpJ* promoter, and this feedback control attenuates the activation of the plant immune system, helping pathogens evade host recognition [29]. Additionally, the transcription of *hrpL* is also affected by other intracellular proteins, such as AefR, CorR, and HrpT, in different *P. syringae* pathovars [12,30,31,32]. These studies suggest that there may also be multiple regulators in *Psa* that regulate the transcription of *hrpR/S* by recognizing different environmental or host signals, but this remains unclear.

Previously, we performed a DNA pull-down assay using the biotin-labeled DNA sequence upstream of *hrpR/S* gene as a probe to detect the protein candidates potentially regulating the transcription of *hrpR/S* gene and found an OmpR-like transcription factor (CN228_13375, GenBank accession: AYL80818.1) as a highly reliable candidate (ProteomeXchange accession: PXD023722) [33]. In this study, the regulatory mechanism and other functions of the OmpR-like transcription factor, involved in regulating the *hrpR/S* gene in *Psa*, were explored by techniques such as gene knockout and overexpression, ChIP-seq, and RNA-seq. The results showed that OmpR-like directly and negatively regulates the transcription of the *hrpR/S* gene and may be involved in the regulation of multiple biological processes in *Psa*. These findings help us understand the function of OmpR-like in *Psa* and provide a basis for elucidating *Psa*’s pathogenic mechanism.

## 2. Results

### 2.1. Overexpression of OmpR-like Resulted in an Attenuated Capacity of Strains to Elicit a Hypersensitive Response in Tobacco Leaves

The strains of *Psa* utilize T3SS to stimulate hypersensitive reaction (HR) production in non-host plants, and the ability to stimulate HR (as measured by the minimum required bacterial concentration) is positively correlated with T3SS gene expression levels [34]. To explore the effect of *OmpR-like* deletion and overexpression on HR induction in tobacco, the wild-type G1, Δ13375 (*OmpR-like* deletion mutant), and OE13375 (an overexpression mutant expressing OmpR-like in Δ13375 under the induction of 1 mM IPTG) were individually injected into tobacco, and leaf necrosis was observed and stained with trypan-blue dye after 24 h. The ability of Δ13375 to stimulate tobacco HR was not significantly different from that of wild-type G1 (Appendix A), while that of OE13375 was significantly decreased (Figure 1a). To examine the expression levels of the T3SS gene in Δ13375 more accurately, we electroporated a previously constructed luciferase reporter plasmid expressing a HrpA-NLuc fusion protein under the control of native *hrpA* promoter, which is induced in T3SS-inducing conditions [35] into G1 and Δ13375. Both strains produced bright fluorescence when injected into non-host *Nicotiana benthamiana* leaves (Figure 1b). We inoculated the strains into King’s B (KB) medium (T3SS inhibition condition) and *hrp*-derepressing medium (HDM) (T3SS induction condition) to detect luciferase activity. In KB, there was no significant difference between luciferase expression in G1 and Δ13375. In HDM, the level of luciferase expression of Δ13375 was slightly but significantly lower than that of G1, indicating that *OmpR-like* deletion reduced the accumulation of HrpA (Figure 1c).

### 2.2. Effects of OmpR-like on T3SS-Related Gene Expression

To explore differentially expressed genes (DEGs) in the *OmpR-like* deletion and overexpression mutants, we performed transcriptome sequencing. The sequencing results showed that the correlation within the sequencing sample group was good (Appendix A), indicating that our transcriptome sequencing results were highly reliable. The Δ13375 strain did not express the transcript of *OmpR-like*, while the value of fragments per kilobase per million mapped reads (FPKM) of this gene in OE13375 was 9.93 times that of the G1 (Figure 2a). Simultaneously, the transcriptome data were also verified by qRT-PCR, and the ORF was not expressed in the deletion mutant, while its expression was significantly increased in the overexpression mutant, as expected (Figure 2b). Compared with G1, Δ13375 displayed 503 upregulated DEGs and 463 downregulated DEGs, while OE13375 showed 934 upregulated DEGs and 767 downregulated DEGs. Additionally, Δ13375 and OE13375 shared 437 DEGs, the former had 529 unique DEGs and the latter had 1264.

Then, we analyzed the transcription levels of 28 genes in the *hrp/hrc* gene cluster (encoding T3SS) in the three strains and found that the expression levels of most genes did not significantly change in Δ13375 but were significantly decreased in OE13375 (Figure 2c). Similarly, 31 type III effector (T3E) genes were analyzed, and the expression levels of most genes did not significantly change in Δ13375, and they decreased significantly in OE13375 (Figure 2d). This suggests that *OmpR-like* overexpression may suppress T3SS and T3E gene expression, while *OmpR-like* deletion has little effect. An analysis of the *hrpR/S* gene found that its transcription level was significantly increased in Δ13375 and significantly decreased in OE13375 (Figure 2c), indicating that OmpR-like may be a negative regulator of the *hrpR/S* gene. Furthermore, unlike other T3E genes, the T3E genes *hopZ5* and *hopH1* (located in the same gene cluster) were downregulated in Δ13375 (Figure 2d), suggesting that OmpR-like may regulate its expression in other ways.

### 2.3. OmpR-like Is a Negative Regulator of the hrpR/S Gene

Transcription factors function mainly through specific binding to genomic DNA. In order to analyze the mechanism of action of OmpR-like transcription factor, we performed chromatin immunoprecipitation sequencing (ChIP-seq). In the ChIP-seq assay, 625 narrow peaks, which were detected in both two ChIP-seq datasets (HDM13375 and KB13375), were used to predict the binding sequence of OmpR-like with MEME-ChIP (https://meme-suite.org/meme/doc/meme-chip.html). The predicted binding sequence of OmpR-like transcription factor was obtained (Figure 3a), and it was found in 327 peaks, 169 of which were distributed upstream of open reading frames (Appendix A). Additionally, the median of the distance between the location of the predicted binding sequence and the transcription start site for these 169 genes was −594 bp. The RNA-seq data revealed 58 and 89 differentially expressed genes (DEGs) bearing such a conserved OmpR-like binding sequence in Δ13375 and OE13375 mutants, respectively (Appendix A). There is a binding site upstream of the *hrpR/S* (Figure 3b) and the expression of *hrpR/S* increased after *OmpR-like* deletion (Figure 3c); the results of qRT-PCR also confirmed those of transcriptome sequencing (Figure 3d), which indicated that OmpR-like is a negative regulator of the *hrpR/S* gene. However, although the *hrpR/S* gene expression was increased in Δ13375 mutant, transcription of its regulon *hrpL* gene was decreased, suggesting a positive effect of OmpR-like on *hrpL* expression, which is independent of the HrpR/S-HrpL regulatory cascade.

### 2.4. OmpR-like Is also Involved in other Psa Biological Pathways

KEGG enrichment of DEGs showed a significant enrichment of functions involving the ribosome (downregulated), bacterial secretion systems (T3SS downregulated, T2SS upregulated, Sec upregulated, and T6SS downregulated), and other regulatory pathways in Δ13375. Biological pathways such as biofilm formation (upregulated), bacterial chemotaxis, bacterial two-component system, flagellar synthesis (upregulated), and bacterial secretion system (T3SS downregulated, T2SS downregulated, Sec upregulated, and T6SS downregulated) were significantly enriched in OE13375, suggesting that OmpR-like is involved in multiple biological pathways (Appendix A).

OmpR-like regulates lipopolysaccharide (LPS) formation by upregulating *lpxC* transcription. Lipopolysaccharide is a component of the outer wall of Gram-negative bacterial cell walls, and lipid A plays a key role in the physiological activity of LPS. According to transcriptome sequencing, both deletion and overexpression of *OmpR-like* can affect LPS synthesis. We determined the LPS content in the wild-type and deletion and overexpression mutants by a phenol-concentrated sulfuric acid method. We found that the LPS formation ability of Δ13375 was attenuated, while OE13375 regained the ability to synthesize LPS (Figure 4a). ChIP-seq showed that OmpR-like has a binding site in the *lpxC* promoter (Figure 4b). Additionally, the expression of the *lpxC* gene was downregulated in Δ13375, whereas it was upregulated in OE13375 (Figure 4c).

Overexpression of *OmpR-like* affects biofilm formation by downregulating *Alg44* (CN228_RS06210). RNA-seq data showed that biofilm synthesis pathways were significantly enriched in the DEGs of OE13375, suggesting that OmpR-like may regulate biofilm formation. Biofilm formation was significantly reduced in OE13375, while it showed no significant difference between Δ13375 and wild-type G1 (Figure 4d). After analyzing the transcription of related genes affecting biofilm pathways in RNA-seq, we identified the decreased expression of genes in the alginate biosynthetic cluster including *Alg44* as a possible reason for this (Figure 4e and Appendix A). Moreover, ChIP-seq peaks detected upstream of the *alg* operon further supported this (Appendix A).

Overexpression of *OmpR-like* results in a decreased swimming motility of strains. Based on transcriptome data, *OmpR-like* is involved in the biological pathway of flagellar assembly in *Psa*. Therefore, we measured the diameter of the strains on KB plates containing 0.3% agar to assess their swimming motility. The deletion of *OmpR-like* had no effect on bacterial swimming motility compared with that in wild-type G1, while *OmpR-like* overexpression significantly reduced it (Figure 4f and Appendix A). Transcriptome data also showed that most genes involved in flagellar assembly were downregulated upon *OmpR-like* overexpression.

Overexpression of *OmpR-like* hinders the growth of strains in KB medium. To investigate the growth of the strains after deletion and overexpression of *OmpR-like*, their growth curves were measured in KB liquid medium, and their OD_600 nm_ value was measured every 3 h. The deletion of *OmpR-like* did not affect strain growth, while *OmpR-like* overexpression decreased it (Figure 4g).

The deletion and overexpression of *OmpR-like* did not affect the pathogenicity of *Psa*. The virulence evaluation of Δ13375 and OE13375 showed that both deletion and overexpression of *OmpR-like* nonsignificantly increased the virulence on leaf discs (Figure 4h and Appendix A). The results of branch wound inoculation showed that, compared with wild-type G1, the deletion and overexpression of *OmpR-like* both nonsignificantly increased the virulence of “Hongyang” and “Guichang” kiwifruit shoots (Figure 4i and Appendix A).

## 3. Discussion

Several studies have considered OmpR-like transcription factors as cognate response regulators of the membrane-bound sensor kinase EnvZ [36,37]. Besides forming a two-component system that controls the expression of outer membrane porins in response to osmotic signals, OmpR and EnvZ also act as global regulators to control the expression of many genes in *Escherichia coli* [38,39]. However, the function of the response regulator OmpR has not been fully understood in *P. syringae*. In this study, the function of the OmpR-like transcription factor in *Psa* was clarified by ChIP-seq, RNA-seq, and gene deletion and overexpression.

First, transcriptome sequencing and qRT-PCR results showed that Δ13375 did not express the transcript of *OmpR-like*, while its expression in OE13375 was 10–20 times that of the wild-type G1, and OmpR-like can be expressed in OE13375 (Appendix A). Secondly, ChIP-seq results showed that OmpR-like transcription factor has a binding site in the promoter region of the *hrpR/S* gene. Transcriptome and qRT-PCR results showed that *OmpR-like* deletion led to an increase in *hrpR/S* transcription, while its overexpression led to a decrease in *hrpR/S* transcription, suggesting that OmpR-like can directly and negatively regulate *hrpR/S* transcription and the expression of downstream T3SS genes. Our DNA pull-down data (ProteomeXchange accession: PXD023722) [33] further supported the direct interaction between the OmpR-like protein and the promoter region of the *hrpR/S* gene. A recent study by Shao et al. [40] also demonstrated that EnvZ-OmpR is a repressor of the T3SS in *P. savastanoi* pv. *phaseolicola* in KB. Actually, the *OmpR-EnvZ* locus is conserved in *P. syringae* complex, and they share over 98% identity between *P. savastanoi* pv. *phaseolicola* (PSPPH_0246/PSPPH_0247) and *Psa* (CN228_01835/CN228_01840), suggesting a similar function of OmpR (CN228_01835) in *Psa.* However, the OmpR-like protein, which is identified in *Psa* in our study, shares only 37% sequence identity with the OmpR protein (PSPPH_0246) described by Shao et al. [40] in *P. savastanoi* pv. *phaseolicola*. Although both the OmpR and OmpR-like proteins are involved in the negative regulation of *hrpR/S* gene expression, they act under different environmental conditions. The OmpR negatively regulated *hrpR/S* under nutrient-rich conditions (e.g., KB medium) but not under T3SS-inducing conditions (e.g., T3SS-inducing minimal medium) [40]. In contrary, we found that the OmpR-like negatively regulated *hrpR/S* gene under T3SS-inducing conditions (HDM) but not in KB medium. Moreover, an EnvZ-like protein, which is encoded by an adjacent gene *CN228_13370* (GenBank accession: AYL80817.1) and shares 28% identity with EnvZ in protein sequence, may serve as the cognate sensor protein of OmpR-like. However, the environmental signals that OmpR-like/EnvZ-like two-component system responds to remain to be determined.

Additionally, KEGG enrichment of DEGs indicated that OmpR-like may be involved in the regulation of multiple biological pathways, such as that of LpxC, which is a positive regulator of key proteins for lipid A synthesis [41]. The expression of *lpxC* was downregulated in Δ13375 and up-regulated in OE13375, suggesting that OmpR-like may affect lipid A synthesis by positively regulating *lpxC* transcription, thereby affecting lipopolysaccharide synthesis. Furthermore, all genes within the alginate synthesis gene cluster showed unaltered expressions in Δ13375 and were significantly downregulated in OE13375, and a binding site of OmpR-like was detected upstream of *alg* operon in the ChIP-seq data, suggesting a negative regulation on biofilm formation by OmpR-like via affecting alginate synthesis. Moreover, as important motility structures, flagella are involved in bacterial motility, chemotaxis, surface attachment, and host cell invasion [42]. In a study by Fu et al. (2022), OmpR upregulated motility by interfering with the expression of flagellar genes, and *ompR* gene deletion in avian pathogenic *Escherichia coli* resulted in a marked inhibition of motility [43]. However, in our study, *OmpR-like* deletion had no effect on bacterial motility compared to that of wild-type G1, whereas *OmpR-like* overexpression significantly reduced bacterial motility. Flagellar synthesis-related genes were upregulated in OE13375, but no binding sites to OmpR-like were observed in ChIP-seq. Therefore, the way in which OmpR-like affects *Psa* motility remains unclear.

Furthermore, many studies have shown that OmpR is directly or indirectly associated with the regulation of virulence genes in pathogens and plays an important regulatory role in pathogenesis [43,44,45,46]. Here, the deletion or overexpression of *OmpR-like* in *Psa* had no significant effect on its pathogenicity against its host kiwifruit. Similarly, a study by Shao et al. (2021) also showed that *ompR* deletion did not significantly change the symptoms and bacterial numbers in bean leaves that underwent infiltration inoculation [40]. This suggests that OmpR and its homologs may not be involved in bacterial pathogenicity in *P. syringae*, though they regulate expression of T3SS genes.

Furthermore, this study is based on a previous nano-luciferase system for accurately monitoring and quantifying T3SS gene expression. For example, changes in T3SS expression levels in Δ13375 were not detected by tobacco HR, but a small but significantly reduced expression of *hrpA*, which is involved in the synthesis of flagella and the regulation of *hrpRS* gene expression [21,47], was found in Δ13375 by the luciferase reporter system. This system can also be used to screen compounds targeting T3SS for the replacement or supplementation of bactericides in disease control. However, given that the reporter plasmids pDSK-hrpA-Nluc and OE13375 both carry resistance to kanamycin, the change of *hrpA* expression in OE13375 was not examined.

Overall, our study clarified the negative regulation of OmpR-like transcription factor on the *hrpR/S* gene, while the reason why OE13375 did not affect pathogenicity remained unclear. Our future work will investigate whether OmpR-like affects the ability of *Psa* to act as an epiphyte on the surface of host leaves. Additionally, the identification of environmental signals (pH, osmolarity, amino acids, or carbohydrates) sensed by the receptor kinase of OmpR-like is also required. The existence of other transcription factors that can interact with OmpR-like in *Psa*, the ways in which these may regulate *hrpR/S* transcription, and the specific regulatory mechanism still need further study.

## 4. Materials and Methods

### 4.1. Strains, Plasmids, Plants, and Growth Conditions

The tested strain *Psa* G1 was isolated from the canker branch of kiwifruit in Xiuwen County, Guiyang City, China. Its genomic data were obtained (GenBank accession: JAESNH01) and used for gene deletion and the construction of luciferase reporter strains. *E. coli* DH5α was used for gene fragment cloning and *E. coli* S17-1λpir was used for conjugative transfer. The suicide plasmid pK18*mobSacB* was used for gene knockout [48]. The expression plasmid pMEKm12 (with ori_1600_ replication origin resulting in 10–15 plasmid copies in *Pseudomonas* and the expression of interest gene under the control of *tac* promoter regulated by the LacI repressor protein) was used to construct the expression vector [49]. The expression plasmid pDSK-GFPuv was used to construct a luciferase reporter strain [50]. Tobacco (*N. benthamiana*) in the four-leaf stage was used to evaluate HR. Three-year-old shoots of kiwifruit (*Actinidia chinensis* var. *chinensis* cultivar ‘Hongyang’ and *A. chinensis* var. *deliciosa* cultivar ‘Guichang’) were used to evaluate the pathogenicity by wound inoculation. The young leaves of ‘Hongyang’ kiwifruit were used to evaluate pathogenicity by leaf disk inoculation. Primers used in this study are listed in Appendix A. *Psa* strains and mutants were grown in LB, KB, or HDM medium at 25 °C. *E. coli* strains were cultivated in LB medium at 37 °C, and the resuscitation of competent cells after electroporation was performed in SOC medium.

### 4.2. Construction of the OmpR-like Deletion Mutant

The number ‘13375’ is specifically refer to the gene encoding OmpR-like (GenBank accession: AYL80818.1; designated as CN228_13375 in the reference genome CP032631.1). The *OmpR-like* deletion mutant of *Psa* G1 was constructed by homologous recombination, based on the suicide plasmid pK18*mobSacB*. Fragments upstream and downstream of *OmpR-like* were amplified by primers 13375 NO/NI and 13375 CI/CO, respectively. The ligated fragment and pK18*mobSacB* were digested with FastDigest restriction enzymes *Bam* HI and *Pst* I, respectively, and the digested fragment and pK18*mobSacB* were ligated with T4 ligase at a molar ratio of 1:3. The ligated product pK18-*13375* was heat-shock transformed into *E. coli* DH5α competent cells and spread on LB-Km50 (LB with 50 μg/mL kanamycin) agar plates. After cultivation at 37 °C for 16 h, clones were detected by primers M13F/PKM13R. Then, the vector was extracted and electroporated (2.5 kv, 5–6 ms) into *E. coli* S17-1λpir. Positive clones were incubated in LB-Km50 liquid medium at 37 °C, and the *Psa* G1 strain was incubated in LB liquid medium at 25 °C to an optical density at 600 nm (OD_600 nm_) of 0.3. Then, 1.5 mL of each suspension was placed in an ice bath for 10 min and resuspended in 400 μL of sterile water after centrifugation. After mixing the two bacterial suspensions in a 1:1 volume ratio, 100 μL of the mixture was placed in the middle of a 0.45 μm sterile Millipore filter placed on an LB plate and incubated at 25 °C for 48 h. The cells on the Millipore filter were resuspended with a suitable LB-K10N10A10 liquid medium (LB with 10 μg/mL kanamycin, 10 μg/mL nalidixic acid, and 10 μg/mL ampicillin) and shaken at 25 °C for 4 h. The 100 μL suspensions were spread onto LB-K10N10A10 agar plates and incubated at 25 °C for 2–3 d. Single colonies were picked and transferred onto LB-Km50 liquid medium for overnight shaking at 25 °C, and primers PsaF/R, SacB-F/R, and 13375-NO/CO were used to detect the transformants. Then, the correct transformants were transferred onto LB liquid medium without NaCl and cultured at 25 °C for 48 h for second recombination. Cultures were diluted 100-fold and spread on LB agar plates (LB without NaCl with 15% sucrose) for 48 h at 25 °C, and primers SacB-F/R, 13375-NO/CO, and 13375-RT-F/R were used to detect the deletion mutant Δ13375.

### 4.3. Construction of the OmpR-like Overexpression Mutant

The *OmpR-like* gene fragment was amplified from *Psa* G1 using primers 13375F/R. The amplified fragment and pMEKm12 were digested with FastDigest restriction enzymes *Pst* I and *Nde* I, respectively, with gel extraction, and then ligated with T4 ligase. The ligation product was heat-shock transformed into *E. coli* DH5α competent cells and spread onto LB-Km50 agar medium at 37 °C for 12 h. Clones were detected by primers M13F/R. The recombinant plasmid was extracted and electroporated (2.5 kv, 5–6 ms) into Δ13375 (*OmpR-like* deletion mutant). Then, 1 mL of SOC medium was used for recovery at 25 °C and 200 rpm for 1 h. Cells were collected, spread onto LB-Km50 plates, and cultured at 25 °C for 48 h. Clones were detected by primers M13F/R and Psa-F/R, resulting in the overexpression mutant OE13375, whose expression is under the control of *tac* promoter and is highly induced by addition of 1 mM of isopropyl L-D-thiogalactoside (IPTG). Positive clones (liquid culture:40% glycerol in a 1:1 ratio) were stored at −80 °C. In the following assays, 1 mM of IPTG was added in bacteria suspensions of OE13375 to induce the expression of OmpR-like protein, and this might not be specially indicated.

### 4.4. Extraction of Bacterial Proteins and Western Blotting

OE13375 (the *OmpR-like* overexpression mutant) was inoculated into 5 mL of LB-Km50 liquid medium supplied with 1 mM IPTG and incubated at 25 °C and 200 rpm for 16 h. Bacteria were collected by centrifugation at 4 °C and 10,000 rpm for 5 min and resuspended in 500 µL of lysis solution (20 mM Tris-HCl, 150 mM NaCl, 10% glycerol, pH 8.0). The resuspension was sonicated for 2 min at 30% power for 4 s at 6 s intervals in an ultrasonic disintegrator (TL-150Y). After sonication, the supernatant was centrifuged at 1000 rpm for 10 min at 4 °C, and its supernatant was transferred to a new centrifuge tube, with the addition of 5 × loading buffer for electrophoretic analysis. The above protein samples were separated by SDS-PAGE and transferred onto PVDF membranes. The membranes were incubated in PBST containing 5% skimmed milk at room temperature for 2 h. The primary antibody (Myc-tag monoclonal antibody, Cat. #: MA121316, Invitrogen, Shanghai, China) was added at a ratio of 1:5000 and incubated at room temperature for 2 h. The membranes were then rinsed three times with PBST for 10 min each time, and the secondary antibody (Cat. #: 31431, Invitrogen, Shanghai, China) was added at a ratio of 1:10,000 and incubated for 1 h. Then, the membranes were washed three times with PBST for 10 min each time. Finally, pictures were obtained with a protein imaging system (Tanon 6600multi, Shanghai, China).

### 4.5. RNA Extraction and qRT-PCR

The tested strains G1, Δ13375, and OE13375 were cultured at 25 °C and 200 rpm for 16 h, centrifuged at 5000 rpm for 5 min, washed with sterile water three times, and suspended in 200 µL of sterile water. The bacterial suspension was added to 1 mL of HDM medium (T3SS induction condition) at a ratio of 1:50 and cultured at 25 °C for 6 h. Samples were treated with RNA protect Bacteria Reagent (Cat. #: 76506, QIAGEN, Shanghai, China), and RNA was then extracted with a bacterial RNA extraction kit (Cat. #: K0731, Thermo Fisher Scientific, Shanghai, China). cDNA was synthesized using a reverse transcription kit (Cat. #: K1691, Thermo Scientific). The qRT-PCR was performed in Bio-Rad CFX96 system using the primers hrpL-F/R and hrpR-F/R, with gyrA-RT-F/R and gyrB-RT-F/R as internal reference primers. The total reaction system was 20 µL. The reaction program was 95 °C for 10 min, followed by 40 cycles of 95 °C for 15 s, 60 °C for 15 s, and 95 °C for 10 min. The results were analyzed using the 2^−ΔΔCT^ method.

### 4.6. Transcriptome Sequencing

The tested strains G1, Δ13375, and OE13375 were cultured at 25 °C and 200 rpm for 16 h, centrifuged at 5000 rpm for 5 min, washed with sterile water three times, and suspended in 200 µL of sterile water. The bacterial suspension was added to 1 mL of HDM medium at a ratio of 1:50 and cultured at 25 °C for 6 h. Samples were treated with the RNA protect Bacteria Reagent (Cat. #: 76506, QIAGEN), and total RNA was then extracted with the bacterial RNA extraction kit (Cat. #: K0731, Thermo Scientific) and sent to a sequencing company (Novogene Co., Ltd., Beijing, China) for transcriptome sequencing. The RNA integrity of samples was assessed using the RNA Nano 6000 Assay Kit of the Bioanalyzer 2100 system (Agilent Technologies, Santa Clara, CA, USA) and mRNA was purified from total RNA using probes to remove rRNA. First strand cDNA was synthesized using random hexamer primer and M-MuLV Reverse Transcriptase, and the residual RNAs were degraded with RNaseH. The second strand of cDNA was synthesized with a Second Strand cDNA Synthesis Kit (Cat. #: A48570, Invitrogen). Then, the Illumina adaptor with hairpin loop structure was ligated to the cDNA, and the resulting fragments that were 370~420 bp in length were selected with AMPure XP system (Beckman Coulter, Beverly, CA, USA). Then, the library was constructed with standard PCR protocol, and its quality was assessed on the Agilent Bioanalyzer 2100 system. The library preparations were sequenced on an Illumina Novaseq platform, and 150 bp paired-end reads were generated. The raw reads were filtered with Trimmomatic v0.36 (Bjoern Usadel, Forschungszentrum Jülich, Leo-Brandt-Straße, Jülich, Germany), and the clean data were mapped to the reference *Psa* genome (GenBank: CP032631.1) with Bowtie2 v2.2.3 (W. B. Langdon, University College London, London, UK). The read counts mapped to each gene were calculated with HTSeq v0.6.1 (Simon Anders, Heidelberg, Germany) and the FPKM (Fragments Per Kilobase of transcript sequence per Millions base pairs) of each gene was calculated based on the length of the gene and reads count mapped. The differentially expressed genes (DEGs) were determined with a DESeq R package v1.18.0 (adjusted *p* value < 0.05) (Simon Anders, Heidelberg, Germany). The Gene Ontology (GO) enrichment analysis of DEGs was implemented by the GOseq R package (Alicia Oshlack, The Walter and Eliza Hall Institute of Medical Research, Parkville, Australia), and the Kyoto Encyclopedia of Genes and Genomes (KEGG) pathways with statistically significant differences were detected with KOBAS v2.0 (Liping Wei, Peking University, Beijing, China).

### 4.7. Chromatin Immunoprecipitation Sequencing (ChIP)

The Pierce^TM^ Magnetic ChIP Kit (Cat. #: 26157, Thermo Fisher Scientific, Shanghai, China) was used for sample pretreatment and immunoprecipitation. The myc-tagged OE13375 was cultured either in HDM or KB broth to an OD_600_
_nm_ of 0.2–0.6, and 1% formaldehyde (final concentration) was added, followed by incubation at 37 °C for 10 min. Then, 125 mM glycine (final concentration) was added and incubated for 5 min at room temperature. Cells were washed twice with cold PBS and suspended in PBS containing protease inhibitors, with suspensions transferred onto new 1.5 mL centrifuge tubes and cells collected by centrifugation. Then, a membrane extraction buffer containing protease/phosphatase inhibitors was used to resuspend the cells and incubate them for 10 min on ice, after which they were collected. Cells were suspended in 200 μL of MNase digestion buffer, and after 15 min in a water bath at 37 °C, 20 μL of MNase stop solution was added to stop the reaction. The chromatin was randomly sheared into fragments smaller than 500 bp by sonication, the output power was 30%, and samples were sonicated for 5 s with 8 s intervals. After centrifugation at 9000× *g* for 5 min, the supernatant was transferred onto a new 1.5 mL tube. Then, 10 μL of the chromatin-containing supernatant, which is 10% of the total input sample of ChIP, was stored at −20 °C, while 90 μL of the supernatant was transferred to 410 μL of dilution buffer. Then, 10 μL of Anti-RNA polymerase II was added to the positive control reaction, 2 μL of normal rabbit IgG was added to the negative control, and 2 μL of mouse monoclonal Myc antibody (Cat. #: MA121316, Invitrogen) was added to the target sample. The above reactions were mixed and incubated at 4 °C for 2 h or overnight. Then, 20 μL of dynabeads was added to each reaction, and the protein and dynabeads were mixed by vortexing and incubated at 4 °C for 2 h. The dynabeads were collected with a magnetic grate, and the supernatant was carefully removed. Subsequently, 1 mL of the eluate was added to wash off unbound proteins, which was repeated three times. Then, 150 μL of eluates was added to the 10% input sample and the IP sample, followed by incubation in a metal bath at 65 °C for 40 min with mixing every 10 min. Dynabeads were collected on a magnetic grate, and the supernatant (containing the eluted protein–chromatin complex) was placed in a centrifuge tube containing 6 μL NaCl (5 M) and 2 μL proteinase K (20 mg/mL), followed by vortexing and incubation at 65 °C for 1.5 h. The obtained samples were purified and sent to a company (Novogene Co., Ltd., Beijing, China) for chromatin immunoprecipitation sequencing. The DNA purity was checked using the NanoPhotometer^®^ spectrophotometer (IMPLEN, Westlake Village, CA, USA), and DNA concentration was measured using Qubit^®^ DNA Assay Kit in Qubit^®^ 2.0 Flurometer (Life Technologies, Carlsbad, CA, USA). The DNA fragments were converted into blunt ends via exonuclease/polymerase activities. After adenylation of 3′ ends of DNA fragments, the NEBNext Adaptor with hairpin loop structure was ligated to prepare for hybridization. In order to select cDNA fragments, preferentially of 150~200 bp in length, the library fragments were purified with AMPure XP system (Beckman Coulter, Beverly, CA, USA). Then, 3 µL of USER Enzyme (Cat. #: M5505S, NEB, Beijing, China) was used with size-selected, adaptor-ligated cDNA at 37 °C for 15 min followed by 5 min at 95 °C before PCR. Then, PCR was performed with Phusion High-Fidelity DNA polymerase (Cat. #: F530L, Thermo Fisher Scientific, Shanghai, China), Illumina Universal PCR primers, and Index Primer. At last, PCR products were purified using AMPure XP system (Beckman Coulter, Beverly, USA), and library quality was assessed on the Agilent Bioanalyzer 2100 system (Agilent Technologies, Santa Clara, CA, USA). Two ChIP-seq datasets, KB13375 and HDM13375, were obtained. The ChIP-seq reads were trimmed from the 3’ end until the final base had a quality score >30 using Fastp v0.19.11(https://github.com/OpenGene/fastp), and then were aligned to the reference genome (GenBank: CP032631.1) using BWA version 0.7.12-r1039. Finally, the peaks were called using MACS v2.1.0 (Y. Zhang, a Python package with open source) with the significance cut-off q-value ≤ 0.05.

### 4.8. Hypersensitive Reaction Assays

Tobacco (*N. benthamiana*) in the four-leaf stage was used to probe the ability of *OmpR-like* deletion and overexpression mutants to induce HR. The bacterial suspensions (with an OD_600 nm_ of 0.2) of tested strains in 10 mM MgCl_2_ and three bacterial suspensions that were serially diluted fivefold were infiltrated into tobacco leaves using a blunt-end plastic syringe, and the diameter of bacterial liquid penetration was about 1 cm. Leaves were observed for the HR within 28 h after inoculation and stained with trypan-blue solution (0.67 mg/mL) for 5 min at 95 °C. Then, 2.5 mg/mL chloral hydrate was used for decolorization, and the decolorization solution was replaced every 2 h. Photographs were obtained after complete decolorization.

### 4.9. Luciferase Assay

The luciferase reporter plasmid was electroporated into competent cells of *Psa* G1 and Δ13375. Positive transformants were inoculated into LB-Km50 broth and cultured at 25 °C and 220 rpm to an OD_600 nm_ of 0.2–0.4. Cells were collected by centrifugation at 1700× *g* at 4 °C for 5 min and gently suspended in cooled sterile water to an OD_600 nm_ of 0.5. The bacterial suspensions of each strain were syringe-infiltrated into the *N. benthamiana* leaves, and luciferase activity was imaged after 6 h using a CCD imaging system (Tanon 6600multi). Then, 50 µL bacterial suspensions of each strain were inoculated into 1 mL of liquid KB (T3SS inhibition medium) and HDM (T3SS induction medium) [51] in 2 mL centrifuge tubes in three replicates, and incubated at 25 °C and 220 rpm for 6 h. Luciferase expression was detected with a chemiluminescence detector (GloMax NAVIGATOR, Promega, Madison, WI, USA).

### 4.10. Lipopolysaccharide Determination

Bacterial suspensions of G1, Δ13375, and OE13375 (each with the same concentration) were individually inoculated into conical flasks containing 50 mL KB medium at a ratio of 1:50 and cultured for 6 h, after which lipopolysaccharides were extracted with a bacterial lipopolysaccharide (LPS) extraction kit (Cat. #: HR8082, Beijing-Biolab Co., Ltd., Beijing, China). Test tubes were numbered 0–6, and 0, 50, 100, 200, 300, 400, and 500 μL standards of 1 mg/mL glucose were added to each tube. Subsequently, 500 μL of 6% phenol and 1.5 mL of concentrated sulfuric acid were added to each tube in turn. Finally, water was added up to 3 mL to allow for a reaction at room temperature for 30 min. The absorbance of the standards at 487 nm was measured using test tube 0 as a blank control. The standard curve was plotted using the sugar content as the horizontal coordinate and the absorbance as the vertical coordinate to find the standard equation. Samples were diluted to arbitrary concentrations, and their absorbance was measured to determine the lipopolysaccharide content from the standard equation.

### 4.11. Biofilm Formation Assay

The tested strains G1, Δ13375, and OE13375 were cultured at 25 °C and 200 rpm for 16 h, centrifuged at 5000 rpm for 5 min, and washed with sterile water three times. Cells were suspended in sterile water to the same OD_600 nm_ value and dropped into an enzyme-labeled plate containing 200 μL KB medium at a ratio of 1:50. The final concentration was adjusted to an OD_600 nm_ of 0.1, and cells were cultured at 25 °C for 3–4 days. Biofilms in the 96-well plates were stained with 0.1% crystal violet stain for 15 min, and the remaining cells adhering to the wells were washed with ddH_2_O three times. The remaining crystal violet was fully dissolved in 100 μL of 95% ethanol. Then, its absorbance value at 590 nm was measured with a microplate reader.

### 4.12. Swimming Motility Assay

The tested strains G1, Δ13375, and OE13375 were cultured by shaking overnight, transferred to 2 mL of KB medium at a ratio of 1:100, and cultured at 25 °C to an OD_600 nm_ of 0.6. Then, 2 μL aliquots were individually dropped onto KB plates containing 0.3% agar and incubated at 25 °C for 24 h. The diameter of the colony was used for quantitative analysis [52]. Experiments were conducted in triplicate for each strain.

### 4.13. Growth Curve Assay

To determine the effect of *OmpR-like* deletion and overexpression on bacterial growth, the tested strains G1, Δ13375, and OE13375 were cultured overnight at 25 °C and 200 rpm, and the bacterial concentration was adjusted to an OD_600 nm_ of 1.0. The bacterial solution was inoculated into a conical flask containing 20 mL of LB broth at a ratio of 1:100 and cultured at 25 °C and 200 rpm. The absorbance at 600 nm was measured every 3 h. Three replicates were set for each sample, and a standard curve was drawn using absorbance values.

### 4.14. Pathogenicity Assessment

For wound inoculation on kiwifruit shoots, the tested strains G1, Δ13375, and OE13375 were cultured at 25 °C and 200 rpm for 16 h, centrifuged at 5000 rpm for 5 min, washed with sterile water three times, and suspended with sterile water to an OD_600 nm_ of 0.2. Three-year-old kiwifruit shoots were cut into about 40 cm in length, disinfected with 0.5% sodium hypochlorite for 20 min, washed with sterile water three times, and dried on a clean bench. Four wounds (2 mm wide and 1 mm deep) were poked on each of the shoots with a blade, and 10 µL of the bacterial solution was dripped onto the wound, with sterile water used as a control. The inoculated shoots were placed in sterile trays lined with moistened filter paper, which were sealed with plastic wrap and then placed in a 16 °C incubator and treated with light for 16 h and dark for 8 h at 95% humidity. The length of the lesions was recorded and photographed after 20 days. For inoculation on kiwifruit leaf discs, healthy leaves were disinfected with 0.5% sodium hypochlorite for 3–5 min, rinsed with sterile water three times, and then dried on a clean bench. The leaves were punched into leaf discs with a 1.1 cm diameter and placed on a 0.5% agar plate (abaxial surface is up). Then, 10 µL of bacterial suspensions was dripped on the center of the leaf discs, with 18 leaf discs inoculated for each strain. These were then placed in a 16 °C incubator and treated with light for 16 h and dark for 8 h at 95% humidity. The area of the lesions was recorded and photographed after 15 days.

### 4.15. Statistical Analysis

Software SPSS 19.0 (IBM, Armonk, NY, USA) was used for data analysis and GraphPad-prism 9.2.0 (GraphPad Software Inc., San Diego, CA, USA) for graphing.

## Figures and Tables

**Figure 1 ijms-23-12306-f001:**
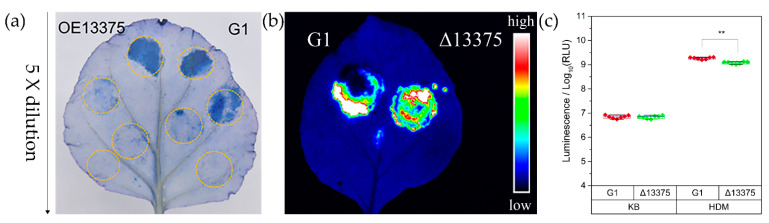
Overexpression of *OmpR-like* resulted in the attenuated capacity of G1 to elicit a hypersensitive reaction in tobacco leaves. (**a**) Strains G1 and OE13375 were individually injected into tobacco (*Nicotiana benthamiana*) leaves. Leaves were stained with trypan-blue dye after 24 h, and the initial bacterial concentration was 5 × 10^7^ CFU/mL, followed by fivefold dilutions. (**b**,**c**) G1 and Δ13375 with a luciferase reporter plasmid both produced bright fluorescence when injected into tobacco leaves, and the level of luciferase expression of Δ13375 was slightly but significantly lower than that of G1 in HDM for 6 h, (**, *p* < 0.01, Student’s *t* test). G1: wild-type strain; Δ13375: *OmpR-like* deletion mutant; OE13375: *OmpR-like* overexpression mutant.

**Figure 2 ijms-23-12306-f002:**
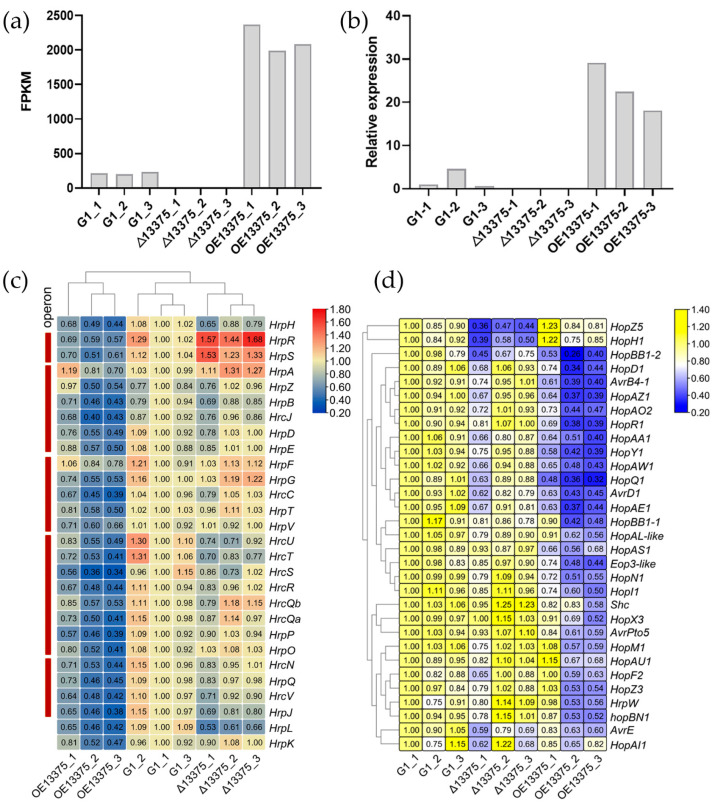
Effects of *OmpR-like* on type III secretion system-related gene expression. (**a**) The transcription levels of *OmpR-like* in G1, Δ13375, and OE13375, based on RNA-seq data. FPKM: fragments per kilobase per million mapped reads; (**b**) The relative expression levels of *OmpR-like* in G1, Δ13375, and OE13375 were detected by qRT-PCR.; (**c**) Effects of *OmpR-like* deletion and overexpression on the transcriptional levels of *hrp*/*hrc* genes in G1, Δ13375, and OE13375; (**d**) Effects of *OmpR-like* deletion and overexpression on the transcriptional levels of type III effector genes in G1, Δ13375, and OE13375. G1: wild-type strain; Δ13375: *OmpR-like* deletion mutant; OE13375: *OmpR-like* overexpression mutant.

**Figure 3 ijms-23-12306-f003:**
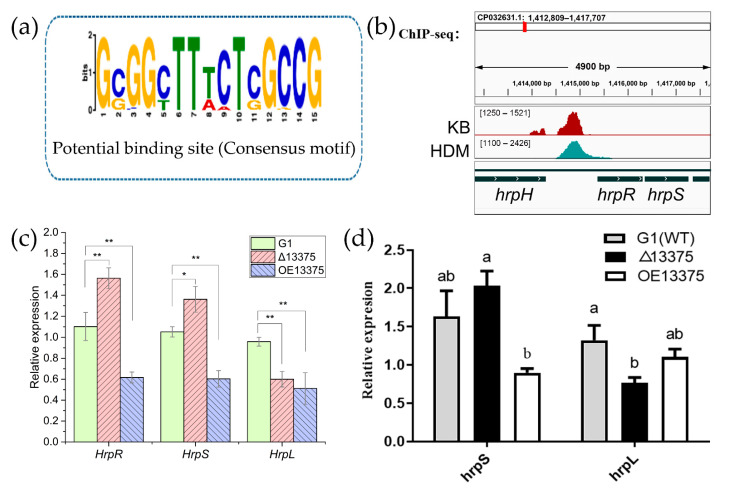
OmpR-like is a negative regulator of the *hrpR/S* gene. (**a**) The most significant motif identified by ChIP-seq using the MEME tool; (**b**) original sequence peaks show the OmpR-like-binding regions in the *hrpR/S* promoter; (**c**) the transcription levels of *hrpR*, *hrpS*, and *hrpL* in G1, Δ13375, and OE13375 were detected by RNA-seq; (**d**) the relative expressions of *hrpS* and *hrpL* in G1, Δ13375, and OE13375 were verified by qRT-PCR. G1: wild-type strain; Δ13375: *OmpR-like* deletion mutant; OE13375: *OmpR-like* overexpression mutant. For two-sample comparison in (**c**), the Student’s *t* test was used (*, *p* < 0.05; **, *p* < 0.01); and different lowercase letters in (**d**) indicate significant differences at *p* < 0.05 level as determined by least significant difference (LSD) tests for multiple comparisons.

**Figure 4 ijms-23-12306-f004:**
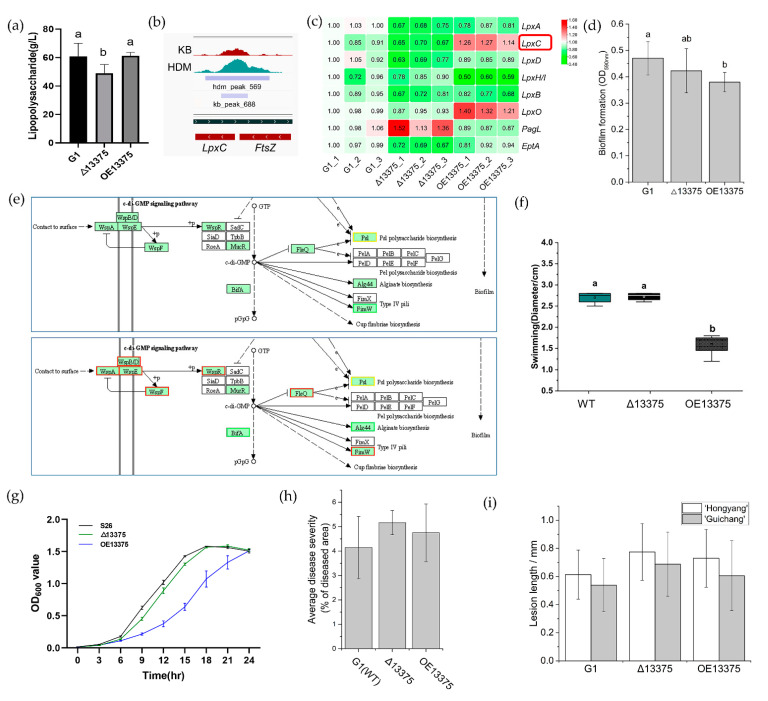
The OmpR-like transcription factor is also involved in other *Psa* biological pathways. (**a**) The lipopolysaccharide content in G1, Δ13375, and OE13375; (**b**) original sequence peaks show the OmpR-like-binding regions in the *lpxC* promoter; (**c**) the differentially expressed genes related to lipopolysaccharide formation in Δ13375 and OE13375 were revealed by RNA-seq analysis; (**d**) the capacity of biofilm formation in G1, Δ13375, and OE13375; (**e**) the transcription of biofilm formation pathway genes in Δ13375 and OE13375; (**f**) swimming motility of G1, Δ13375, and OE13375 and diameter measurements of halos obtained when strains with the same OD_600 nm_ values were titrated on KB plates containing 0.3% agar and incubated within 24 h at 25 °C; (**g**) standard curves of G1, Δ13375, and OE13375 in LB medium were drawn by measuring the optical density at 600 nm of the bacterial suspension every 3 h; (**h**) virulence quantification of G1, Δ13375, and OE13375, obtained by calculating the average disease severity (ratio of diseased area to the total area of the leaf disc); (**i**) pathogenicity of G1, Δ13375, and OE13375 on woody canes of different kiwifruit varieties (“Hongyang” and “Guichang”), determined by the wound-inoculation method. The lesion length was measured 20 days after inoculation. G1: wild-type strain; Δ13375: *OmpR-like* deletion mutant; OE13375: *OmpR-like* overexpression mutant. The difference among samples were determined by least significant difference (LSD) tests for multiple comparisons, and different lowercase letters in indicate significant differences at *p* < 0.05.

## Data Availability

All original articles used to support the findings of this study are available from the corresponding author upon request. The ChIP-seq and RNA-seq data were deposited in Gene Expression Omnibus and Sequence Read Archive database (PRJNA871803).

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
