# Peer review of "The OmpR-like Transcription Factor as a Negative Regulator of hrpR/S in Pseudomonas syringae pv. actinidiae"

_ijms, 2022, doi:10.3390/ijms232012306_

Round 1
Reviewer 1 Report
The Manuscript entitled "The OmpR-like transcription factor as a novel negative regulator of hrpR/S in Pseudomonas syringae pv. actinidiae" by Zao et al. describes the role of OmpR-like transcription factor in the regulation of hrpR/S. Indeed, HrpR/S-HrpL cascade sophisticatedly regulates the expression of T3SS and effectors of T3SS in Pseudomonas syringae pv. actinidiae. From my view point, OmpR as "novel negative regulator of hrpR/S " is not correct. It has been previously published by Shao et al., 2021 and even verified by qRT-PCR that in KB, the expression of hrpR, hrpL, hrpA2, hrpZ1, and hrpK1 was significantly upregulated in the ompR deletion strain and recovered in the complemented strain compared with that in the wild-type. Moreover the role of EnvZ in EnvZ-OmpR two component system has been not discussed in this Manuscript. Moreover, there is no accession number for the ChIP-seq and RNA-seq data submission to Gene Expression Omnibus. Moreover the experimental/methodological details have been poorly described in this paper: strain Δ13375 is an inframe gene deletion strain, OE13375 is not only OmpR-like overexpression mutant but also a complementation strain for ompR-like gene deletion. Genetic organization of hrpR/S and other related genes and putative OmpR-like binding site(s) is not depicted as Figure. Also no detail is provided about the strain OE1337 OmpR-like: how is it expressed in trans, copy number of the plasmid, induction and overexpression conditions of OmpR-like is not at all mentioned. Line 96: luciferase reporter plasmid, based on our previously constructed nano-luciferase system- nothing mentioned about transcription fusion, promoter or induction conditions (if any). Line 157-160: Nothing about test of significance mentioned in the legend of the Figure. The original Western-blot gel image provided corresponding to Fig. S1 in supplementary is without any labels or information.
Reviewer 2 Report
There needs to be an expansion in the Introduction of why and how the OmpR gene was selected for this study. In the abstract it states that the protein was identified using pull-downs, but there is no description of this in the Methods or Results. As a consequence, it is very difficult to determine which protein/gene is the subject of this study.
There is a general confusion in the manuscript around the nomenclature for the genes referred to in this study. For example this is in the Introduction:
'In this study, the regulatory mechanism and other functions of the OmpR-like transcription factor (GenBank: CP032631.1), involved in regulating the hrpR/S gene in Psa' The GenBank number referred to CP032631.1 is the complete sequence of Psa M228, and not an identifier for OmpR.
The Psa strain used in this study is G1. It has apparently been sequenced but it needs to be deposited in GenBank or other public depository so it can be accessed.
These need to be done before the paper can be published:
An explanation of how OmpR was identified and cloned needs to be provided;
The sequence of G1 needs to be submitted to GenBank or other public database;
There needs to be clarity around the identity of the gene numbering used in this study.
Round 2
Reviewer 1 Report
Line 159: "The predicted binding sequence" was upstream of hrpR/S.
How many of such conserved sequences are distributed in the genome upstream of other open reading frames? When present, how far upstream are they usually located relative to respective transcription start point of the gene? how was the consensus derived? How many from the modulated genes detected by RNA-seq in Δ13375 strain bear such conserved OmpR-like binding sequence(s)?
Line 164-167: "However, although the hrpR/S gene expression was increased in Δ13375 mutant, transcription of its regulon hrpL gene was decreased, suggesting a positive effect of OmpR-like on hrpL expression which is independent of HrpR/S-HrpL regulatory cascade":
From RNA-seq analysis how many other genes are positively regulated by OmpR-like? It seems there it is acting like a transcription activator. qRT-PCR might be carried out to substantiate RNA-seq results on the expression of some of the genes, outside the hrp genes, like lpx, motility related candidate gene(s) etc which have been speculated to be positively regulated by OmpR-like.
line 216: "The deletion and overexpression of OmpR-like did not affect the pathogenicity": As it has been also discussed that, "more sensitive methods, such as bacterial counts, may be needed to detect the effect of OmpR-like deletion or overexpression on pathogenicity of Psa", I think that this result is not convincing enough to be presented in the manuscript.
line 79: upstream and not upstreaming.
line 289: date of reference
line 301: date of reference
Round 3
Reviewer 1 Report
Manuscript well improved